# Impact of Construction and Functioning of a Newly Built Ski Slope on the Quality of Nearby Stream Water

Anna Lenart-Boroń [1],* , Anna Bojarczuk [2] and Mirosław Żelazny [2]

1   Department of Microbiology and Biomonitoring, University of Agriculture in Cracow,
    Mickiewicza Ave. 24/28, 30-059 Cracow, Poland
2   Department of Hydrology, Institute of Geography and Spatial Management,
    Jagiellonian University in Cracow, Gronostajowa 7, 30-387 Cracow, Poland
*   Correspondence: anna.lenart-boron@urk.edu.pl

**Featured Application: This case study can be referred to by the operators and designers of ski slopes as an example of the potential impact that the construction and functioning of a ski slope have on the aquatic environment.**

**Abstract:** The construction of new, artificially snowed, ski slopes and the accompanying infrastructure changes the natural environment and exerts pressure on the ecosystems. This study examined the impact of the construction and operation of a new ski slope, with its infrastructure and artificial snow production, on the quality of nearby stream waters. The research period covered the time before, during and after the slope construction. Electrolytic conductivity (EC) and pH were measured on-site, chemical analyses included the determination of $Ca^{2+}$, $Mg^{2+}$, $Na^+$, $K^+$, $HCO_3^-$, $SO_4^{2-}$, $Cl^-$, $NO_3^-$, and microbiological analysis comprised mesophilic and psychrophilic bacteria, total and fecal coliforms, and *E. coli*. As a result of intensive environmental transformations, the examined parameters varied significantly over the study period, as shown by the coefficient of variation. Due to land cover changes, concentrations of all the examined parameters increased during the ski slope construction due to ions and bacteria leaching from the soil. However, when construction works were finished, all bacterial and some chemical indicators returned to the state observed before the construction, most probably due to the recovery of vegetation and self-purification of water. Supply of melt water from artificial snow, produced from water containing higher concentrations of ions, increased pH, EC, $Ca^{2+}$, $Mg^{2+}$ and $HCO_3^-$ in the stream. Providing that the development of ski stations is unavoidable in the considered region, conducting studies assessing the impact of new ski slope construction is an important step that should be conducted prior to undertaking such investments.

**Keywords:** ski resort; ski slope; snowmaking; stream water; water quality

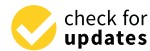



## 1. Introduction

Tourism is considered one of the largest industries worldwide, among which tourism and recreation activities based on natural resources, such as downhill skiing, are of considerable importance [1]. Many ski resorts offer a wide range of après-ski activities and other types of recreation to increase the number of tourists visiting the area, further increasing the construction of the related infrastructure. This requires changing the landscape structure and functions of ecosystems. Moreover, management practices of ski resorts, such as the production of artificial snow, salting and snow grooming, as well as everyday operation of the related infrastructure, such as bars and hotels with their toilets, all may have negative impacts on freshwater systems. According to Pickering et al. [2], winter tourism has a range of impacts on water quality in the surrounding environment with the most important factor being untreated waste discharged to streams and rivers downstream of ski resorts.



The reduced quality of water within ski resorts results from contamination of rivers and streams by nutrients and microorganisms from treated human waste but also from runoff from ski slopes, roads, car parks, etc. [2].

Undoubtedly, the functioning of ski areas affects the natural environment in a wide range of aspects. Construction of ski-related facilities requires the removal of native vegetation and top soil, therefore increasing erosion, surface runoff and nutrient leaching [1]. Milne et al. [3], in a case study in Vermont, attributed increased runoff in the area to forest harvesting, increased soil compaction and snow redistribution. The same authors reported elevated concentrations of sodium and chloride in nearby stream waters as the result of using de-icing road salt on the access road to the ski resort. Additionally, Molles and Gosz [4] in their study on the effect of the Santa Fe ski area on the water quality, found that the concentrations of every major nutrient except $SO_4$ were increased below the ski area as a result of using road salt. They also reported increased erosion and sediment transport. Another negative aspect of the functioning of ski resorts is that they mostly become four-season facilities, therefore water demand is also elevated during the entire year [3]. However, water consumption by residential and resort facilities may be small compared to the demand for snowmaking in winter, which is particularly dangerous as, in winter, water resources are generally lower, therefore baseflow becomes more severe. The production of artificial snow may also have a number of other effects on the water ecosystems resulting in, e.g., different properties of artificial snow compared to the natural snow [5]. Artificial snowmaking increases the amount of water in the watershed, increasing the peak flow and suspended sediment yield [6]. Among the effects of artificial snow production, there is a prolongation of the snow cover, consequently leading to a shortening of the growing cycle or higher ion concentration in artificial snow than in natural snow, since the water used for its production is most often drawn from rivers [7]. Jones and Devarenness [7], when examining the effect of artificial snow on the germination of mountain flora, observed that the chemical composition of artificial snow differed largely from the composition of natural snow. Additionally, Rixen et al. [8] demonstrated significant differences in the chemical composition of artificial snow (most often produced from water taken from rivers) and natural snow. They also claim that higher concentrations of ions in melt water from artificial snow may affect, for example, plant growth. Moreover, there are some concerns raised by the common use of additives (e.g., ice nucleation active strains of *Pseudomonas syringae* or ammonium nitrate) in order to optimize freezing of water related to potential health risks such as allergies for, e.g., snowmakers or the use of poor water quality to make artificial snow [9].

The number of ski resorts and ski runs in Poland has rapidly increased in the past few years and Białka Tatrzańska was rated as the best ski resort in Poland since 2012 [10,11]. The popularity of this region results from favorable natural conditions (climate and landscape), culture and constantly expanding infrastructure, including newly constructed ski runs and ski slopes. The calculations conducted by Krzesiwo [12] suggest that in the winter season of 2014/15, only the Kotelnica Białczańska Ski Resort was visited by about 318,000–342,000 people, of which about 27,000–30,000 were tourists from abroad. However, too high temperatures and too scarce amount of snow are the most important barriers to the development of skiing in Poland, which is why snow cannons, pipelines and retention reservoirs are among the most significant parts of the ski tourism-related infrastructure, as they at least to some extent allow the resorts to become independent of snowfall [10]. The construction and operation of new ski slopes require conducting extensive work and—after launching the ski slope—production of artificial snow, which can be potentially aggravating for the natural environment and surface waters. Such impact, however, has not been examined in detail. Fidelus-Orzechowska et al. [13] conducted a thorough study aimed at assessing the effect of the construction of the considered ski runs on the quantitative and spatial changes in land relief. They found that as a result of the construction of the ski run the catchment elevation was lowered by 0.02 m and they observed changes in the surface runoff patterns in the form of new geometrically regular convergence and divergence zones,

as a result of the construction of new drainage ditches. On the other hand, maintaining good quality and conducting both physico-chemical and microbiological analyses of stream water in mountain areas is important for social and economic reasons, as this affects the attractiveness of tourist areas [14]. Therefore, the aim of this study was to assess whether the construction and exploitation of a ski slope, its infrastructure and the associated artificial snow production, had an impact on the aquatic environment in its vicinity. This was achieved by the determination of changes in physico-chemical parameters and the concentration of selected bacteriological indicators of water quality of a Remiaszów stream, which drains the direct vicinity of the ski slope.

## 2. Materials and Methods

### 2.1. Study Area

The study was conducted on a Remiaszów stream, situated in the catchment of the Białka river, in southern Poland (Figure 1). The catchment of the Remiaszów stream is 1.16 km$^2$ [13]. The ski station is located in Białka Tatrzańska, one of the largest ski centers in Poland. The ski station has nine ski lifts with a transport capacity of 16,775 people per hour [12]. In addition to a well-developed ski base, Białka Tatrzańska has strong and continuously expanding gastronomic, recreational and accommodation infrastructure and the transport capacity of all ski lifts used in the Białka Tatrzańska town is 28,395 people per hour [15].

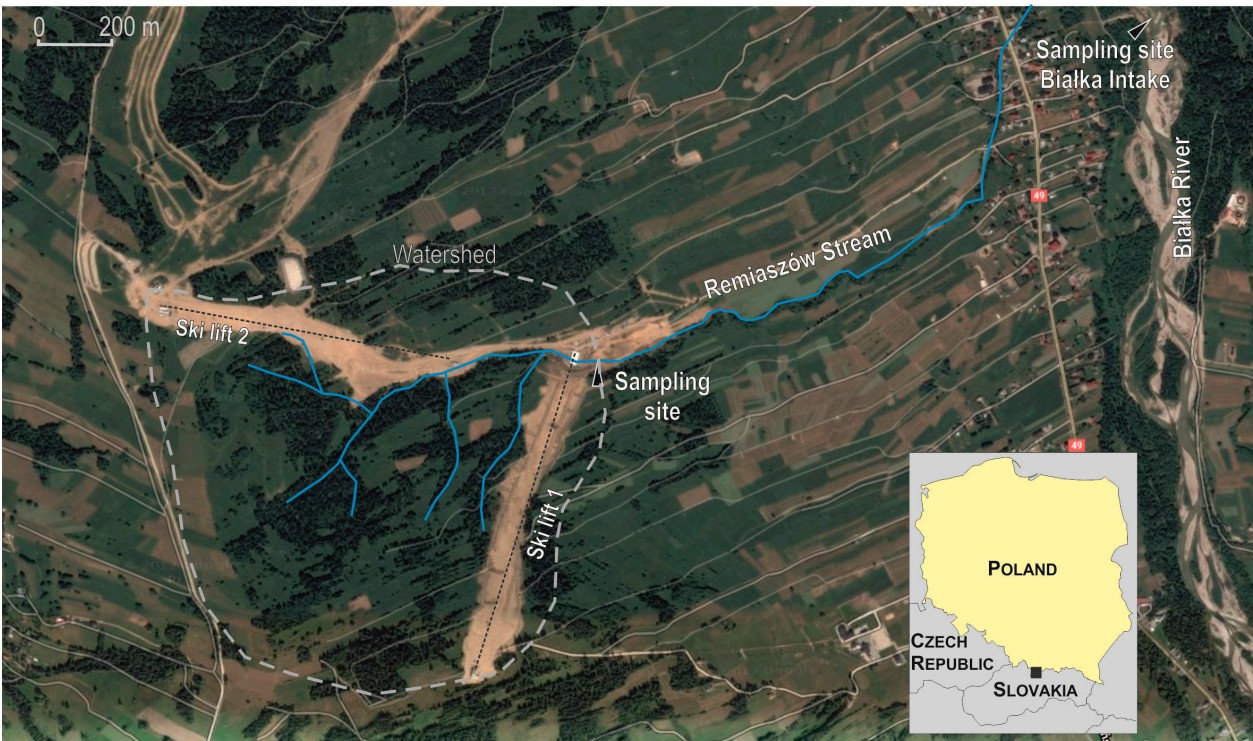

**Figure 1.** Study area.

The ski resort in Białka Tatrzańska is being constantly modernized and expanded. Two of the newest ski runs were built along the Remiaszów stream. The construction of the newest ski run (Remiaszów, No. 2 in Figure 1) began in early spring 2015 and the slope was opened to the skiers at the end of December 2015. Both slopes are approx. 900 m long and they cover a total area of 0.195 km$^2$, which constitutes 16.8% of the catchment area [13].

During the construction of the slope, it was necessary to cut out fragments of the nearby forest, and earthworks were needed for profiling the ski slopes. Apart from the ski infrastructure (lift stations, supports, artificial reservoirs), other facilities were also created, such as bars, toilets, car parks, access roads, etc. This slope is artificially snowed. Water

for the production of artificial snow is drawn from a retention reservoir with a capacity of 50,000 m$^3$ supplied from the river Białka, which flows nearby the ski resort. In summer, the area of the ski station is used for sheep grazing.

### 2.2. Sampling Strategy

The study was carried out from August 2014 to March 2017. The samples were collected in 21 sampling series, conducted on a monthly or two-week basis during ski seasons (from early December to April) and every second month in summer and autumn. Field measurements of electrolytic conductivity (EC) and water pH in the Remiaszów stream and in the Białka river were carried out using a Pro 2030 Multimeter handheld (YSI, US). For chemical analyses, water samples from the Remiaszów stream and the Białka river were collected into 500 mL polyethylene bottles, while for microbiological analyses—into 1000 mL polypropylene sterile bottles. Samples of water from atmospheric precipitation were also collected from the fluviometer installed at the area of the ski station, but only physico-chemical characteristics were determined therein.

### 2.3. Laboratory Analyses

The concentration of the following ions: $Ca^{2+}$, $Mg^{2+}$, $Na^+$, $K^+$, $NH_4^+$, $HCO_3^-$, $SO_4^{2-}$, $Cl^-$, $NO_3^-$, $NO_2^-$ and $PO_4^{3-}$ was determined in the Hydrological and Chemical Laboratory of the Institute of Geography and Spatial Management, Jagiellonian University by ion chromatography (DIONEX ICS-2000) and mineralization of water (TDS) was calculated as the sum of the determined ions. $NH_4^+$, $NO_2^-$, $PO_4^{3-}$ were not included in further statistical analyses due to their very low values in the Remiaszów stream, usually below limit of detection.

The number of bacteriological indicators of water quality was assessed in the microbiological laboratory of the University of Agriculture. Serial dilutions method was used to assess the number of mesophilic and psychrophilic bacteria (Trypticase Soy Agar, 48 h at 37 °C and 72 h at 4 °C, respectively). Membrane filtration was used to enumerate total and fecal coliforms (purple red colonies with metallic sheen on Endo agar, incubation at 37 °C and 44 °C, respectively for 48 h), as well as total and fecal *Escherichia coli* (blue–green colonies on TBX agar, incubation at 37 °C and 44 °C, respectively, for 48 h). After incubation, visible colonies were counted and expressed as colony-forming units per 100 mL in the case of membrane filtration method and per 1 mL in the case of serial dilutions method (CFU/100 mL and CFU/mL).

### 2.4. Statistical Analysis

In order to assess whether there are statistically significant differences in the values of microbiological indicators of water quality and hydrochemical parameters of water in the Remiaszów stream before, during construction and during operation of the ski slope, ANOVA and post-hoc Least Significant Difference (LSD) tests were conducted for *p* at the significance level of 0.05

## 3. Results

### 3.1. Chemical Composition of Precipitation Water

Precipitation water in the studied area is characterized by low conductivity, mineralization and its pH is slightly acidic (Table 1). The chemical composition is dominated by $Ca^{2+}$ among cations and by $HCO_3^-$ among anions. Sulfates and nitrogen compounds also have quite a large share in the chemical composition of precipitation water.

**Table 1.** Physico-chemical characteristics of precipitation water and basic statistics.

| Feature | Unit | Mean | Median | Min | Max | Q25 * | Q75 * | CV * [%] |
|---|---|---|---|---|---|---|---|---|
| pH | – | 6.34 | 6.42 | 5.56 | 6.85 | 6.09 | 6.67 | 7.3 |
| EC | μS/cm | 31.27 | 32.35 | 17.90 | 46.90 | 20.60 | 37.50 | 34.9 |
| TDS | mg/L | 21.37 | 23.20 | 12.01 | 31.63 | 12.61 | 25.56 | 36.1 |
| $Ca^{2+}$ | | 3.478 | 4.052 | 1.028 | 5.475 | 1.560 | 4.703 | 51.4 |
| $Mg^{2+}$ | | 0.463 | 0.490 | 0.116 | 0.648 | 0.418 | 0.618 | 42.0 |
| $Na^+$ | | 0.435 | 0.429 | 0.249 | 0.572 | 0.405 | 0.525 | 25.7 |
| $K^+$ | | 0.689 | 0.466 | 0.233 | 1.724 | 0.369 | 0.879 | 80.1 |
| $NH_4^+$ | | 0.634 | 0.442 | 0.266 | 1.756 | 0.385 | 0.514 | 87.6 |
| $HCO_3^-$ | | 8.431 | 8.125 | 2.438 | 12.780 | 7.038 | 12.082 | 44.7 |
| $SO_4^{2-}$ | | 3.848 | 4.107 | 1.947 | 5.636 | 2.413 | 4.879 | 36.9 |
| $Cl^-$ | | 1.000 | 0.969 | 0.323 | 1.704 | 0.876 | 1.156 | 44.9 |
| $NO_3^-$ | | 2.306 | 1.992 | 1.189 | 4.288 | 1.576 | 2.802 | 48.2 |
| $NO_2^-$ | | 0.078 | 0.085 | 0.033 | 0.127 | 0.034 | 0.104 | 49.4 |

* Q25—lower quartile, 25% of data are below this value; Q75—upper quartile, 25% of data are above this value. CV—coefficient of variation.

### 3.2. Quality of Water Used for the Production of Artificial Snow

In contrast to the precipitation water, water drawn from the river Białka, used for the production of artificial snow on the examined ski slope, is characterized by about 10 times higher mineralization and alkaline pH (Table 2). The chemical composition is also dominated by calcium and bicarbonates but the concentration of $Ca^{2+}$ ions is 10 times higher, while the concentration of $HCO_3^-$ is 15 times higher. On the other hand, the concentrations of $NO_3^-$ in water used for artificial snow production are very similar to rainwater. Numerous coliforms, fecal coliforms and *E. coli* were found in the river Białka, evidencing the contamination of water by human and animal waste.

**Table 2.** Physico-chemical characteristics, bacteriological indicators of water quality and basic statistics—river Białka, water intake for artificial snow production.

| Feature | Unit | Mean | Median | Min | Max | Q25 * | Q75 * | CV * [%] |
|---|---|---|---|---|---|---|---|---|
| pH | – | 7.66 | 7.74 | 7.03 | 8.15 | 7.42 | 7.94 | 4.6 |
| EC | μS/cm | 250.4 | 256.5 | 133.7 | 299.6 | 251.3 | 269.6 | 16.2 |
| TDS | mg/L | 197.2 | 202.4 | 100.9 | 230.1 | 197.1 | 208.6 | 16.6 |
| $Ca^{2+}$ | | 36.1 | 36.8 | 19.5 | 42.8 | 35.6 | 38.7 | 16.1 |
| $Mg^{2+}$ | | 7.5 | 7.7 | 4.0 | 8.9 | 7.4 | 8.1 | 16.4 |
| $Na^+$ | | 3.87 | 3.85 | 1.39 | 5.27 | 3.45 | 4.63 | 28.0 |
| $K^+$ | | 0.832 | 0.835 | 0.629 | 1.057 | 0.761 | 0.909 | 13.9 |
| $HCO_3^-$ | | 117.5 | 122.6 | 59.9 | 141.3 | 115.0 | 124.8 | 16.9 |
| $SO_4^{2-}$ | | 22.4 | 23.4 | 10.4 | 26.8 | 21.4 | 25.1 | 19.6 |
| $Cl^-$ | | 5.2 | 5.3 | 1.7 | 7.7 | 4.3 | 6.6 | 33.7 |
| $NO_3^-$ | | 3.5 | 3.5 | 2.7 | 4.8 | 3.1 | 3.7 | 16.8 |
| Total coliforms | CFU/100 mL | 14,980 | 4390 | 49 | 112,000 | 106 | 13,000 | 209.6 |
| Total *E. coli* | | 6570 | 925 | 5 | 44,600 | 80 | 5100 | 202.6 |
| Fecal coliforms | | 3350 | 220 | 30 | 21,000 | 105 | 2195 | 201.7 |
| Fecal *E. coli* | | 2600 | 130 | 21 | 19,500 | 51 | 1770 | 221.0 |
| Mesophilic bacteria. | CFU/mL | 3270 | 720 | 12 | 14,510 | 263 | 4198 | 153.4 |
| Psychrophilic bacteria. | | 3450 | 1950 | 30 | 9000 | 740 | 6050 | 93.1 |

* Q25—lower quartile, 25% of data are below this value; Q75—upper quartile, 25% of data are above this value. CV—coefficient of variation.

### 3.3. Chemical Composition and Microbiological Quality of Water in the Remiaszów Stream

The water in the Remiaszów stream has alkaline pH and the mean mineralization is c.a. 280 mg/L (Table 3). The chemical composition is dominated by $HCO_3^-$ among anions and by $Ca^{2+}$ among cations. $Mg^{2+}$ and $SO_4^{2-}$ have slightly lower values. The concentration of remaining ions generally does not exceed 10 mg/L. Mesophilic and psychrophilic bacteria were always found in these samples of water. The number of coliforms and fecal coliforms

in the water of this stream is highly variable, as evidenced by high coefficients of variation (CV > 200%). However, it can be observed that the number of bacteria in the Remiaszów stream is definitely lower than in water drawn from the Białka river, used for the production of artificial snow.

**Table 3.** Physico-chemical characteristics, bacteriological indicators of water quality and basic statistics—Remiaszów stream.

| Feature | Unit | Mean | Median | Min | Max | Q25 * | Q75 * | CV * [%] |
|---|---|---|---|---|---|---|---|---|
| pH | − | 7.8 | 7.9 | 7.1 | 8.2 | 7.5 | 8.1 | 4.2 |
| EC | µS/cm | 328.8 | 338.1 | 201.5 | 387.1 | 297.9 | 365.9 | 14.4 |
| TDS | mg/L | 280.9 | 282.7 | 157.7 | 335.2 | 254.1 | 323.4 | 16.3 |
| $Ca^{2+}$ | | 54.2 | 54.6 | 33.2 | 62.9 | 49.5 | 59.9 | 13.7 |
| $Mg^{2+}$ | | 8.53 | 8.69 | 4.49 | 12.06 | 7.03 | 10.08 | 21.5 |
| $Na^+$ | | 3.27 | 2.72 | 1.64 | 10.17 | 2.43 | 3.08 | 54.1 |
| $K^+$ | | 1.02 | 0.98 | 0.73 | 1.55 | 0.92 | 1.06 | 20.0 |
| $HCO_3^-$ | | 186.8 | 187.0 | 101.0 | 237.6 | 161.9 | 214.9 | 18.7 |
| $SO_4^{2-}$ | | 21.4 | 21.4 | 11.4 | 35.3 | 19.1 | 24.4 | 24.1 |
| $Cl^-$ | | 3.08 | 2.09 | 1.13 | 16.20 | 1.43 | 2.73 | 105.4 |
| $NO_3^-$ | | 2.46 | 1.85 | 0.81 | 6.02 | 1.38 | 3.51 | 58.5 |
| Total coliforms | CFU/100 mL | 838 | 80 | 0 | 7700 | 1 | 1000 | 209.8 |
| Total *E. coli* | | 572 | 20 | 0 | 7200 | 0 | 400 | 270.0 |
| Fecal coliforms | | 473 | 8 | 0 | 6900 | 0 | 170 | 305.4 |
| Fecal *E. coli* | | 508 | 6 | 0 | 5620 | 0 | 550 | 253.3 |
| Mesophilic bacteria. | CFU/mL | 737 | 310 | 15 | 4015 | 60 | 780 | 150.8 |
| Psychrophilic bacteria. | | 10,404 | 3260 | 40 | 88,000 | 420 | 8600 | 211.6 |

* Q25—lower quartile, 25% of data are below this value; Q75—upper quartile, 25% of data are above this value. CV—coefficient of variation.

*3.4. The Impact of the Construction and Operation of a Ski Run on the Quality of Water in the Remiaszów Stream*

As shown by the results of ANOVA, the quality of water in the Remiaszów stream during the construction of the ski run differed significantly from the periods before its construction and during its operation in terms of the following parameters: pH, $K^+$, total *E. coli*, fecal coliforms, fecal *E. coli*. On the other hand, the values of $NO_3^-$ in the water differed significantly between the construction period and the period after the construction—during the operation of the ski run (Table 4, Supplementary data). A statistically significant decrease in the pH value of the stream water and increased values of $K^+$, total *E. coli*, fecal coliforms and fecal *E. coli* were observed during the construction of the ski run. However, in the later period, the values of these parameters returned to a state similar to those recorded before the ski run construction. In the case of $NO_3^-$, the concentrations were the highest during construction, while during the operation of the newly built ski run, they decreased and were even lower than the ones recorded before the construction works began (Figure 2).

In the case of the remaining parameters of the stream water, there were no significant differences between the periods before, during and after the construction of the considered ski slope. However, in the case of EC, TDS, $Ca^{2+}$, $Mg^{2+}$, $HCO_3^-$, $SO_4^{2-}$, total coliforms and mesophilic bacteria, a slight increase was observed in the period after the construction, compared to the period before the construction began (Figure 2). On the other hand, the concentrations of $Na^+$, $Cl^-$ and the number of psychrophilic bacteria slightly decreased in the stream water.

**Table 4.** ANOVA results showing differences in physico-chemical characteristics and bacteriological indicators of water quality (Remiaszów stream) between different sampling periods.

| Feature | F * | p ** |
|---|---|---|
| pH | 16.645 | **0.000** |
| EC | 0.793 | 0.466 |
| TDS | 0.460 | 0.638 |
| $Ca^{2+}$ | 0.475 | 0.628 |
| $Mg^{2+}$ | 0.301 | 0.743 |
| $Na^+$ | 0.493 | 0.618 |
| $K^+$ | 8.740 | **0.002** |
| $HCO_3^-$ | 0.718 | 0.500 |
| $SO_4^{2-}$ | 0.376 | 0.691 |
| $Cl^-$ | 0.960 | 0.400 |
| $NO_3^-$ | 3.627 | **0.045** |
| Total coliforms | 1.191 | 0.325 |
| Total *E. coli* | 4.927 | **0.018** |
| Fecal coliforms | 11.008 | **0.001** |
| Fecal *E. coli* | 8.402 | **0.002** |
| Mesophilic bacteria | 2.614 | 0.098 |
| Psychrophilic bacteria | 0.466 | 0.634 |

* F—the critical value of ANOVA. ** Probabilities (p) at which differences are considered significant are set in boldface.

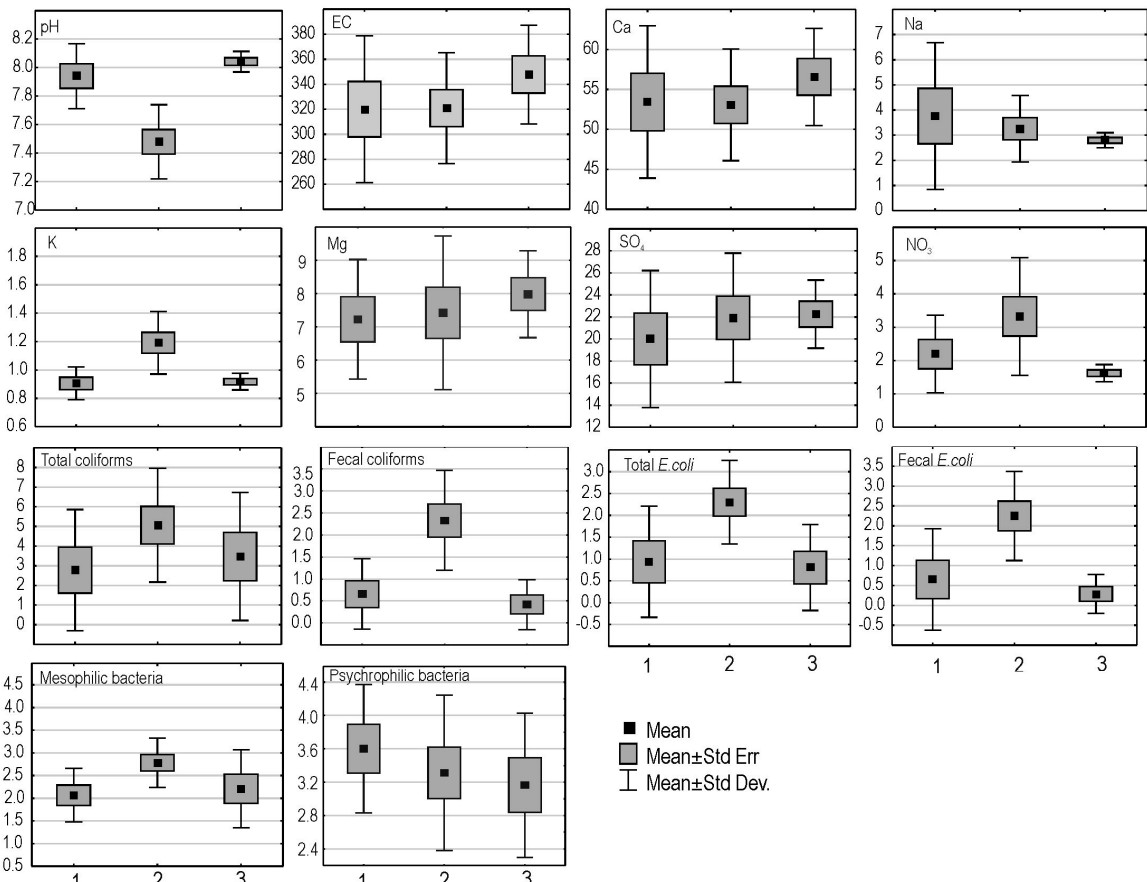

**Figure 2.** Values of pH and EC [μS/cm], concentration of ions [mg/L], number of bacteria [logCFU/100 mL and logCFU/mL] in water of the Remiaszów stream: before construction (1), during construction (2) and during operation (3) of the considered ski slope.

## 4. Discussion

Mountain streams provide a habitat for aquatic animals and clean water for human consumption downstream. The stream water passes the adjacent terrestrial ecosystem which largely affects the water quality [16]. The condition of mountain streams reflects the processes and activities occurring within the upstream part of the watershed.

Before the construction of the new ski slope, atmospheric precipitation was among the predominant factors affecting the chemical composition of the Remiaszów stream [17]. The presence of calcium and magnesium in rainwater is a factor aiding the neutralization of acid rain [18]. In general, the chemical composition of precipitation water is largely formed by dissolving particulate matter in the atmosphere [17]. Regardless of the predominant ions in the rainwater, it needs to be stressed that it was characterized by low levels of all chemical compounds, as evidenced by low TDS value. This suggests that after the construction of the ski slope, increased concentrations of some ions in the water of this stream could be expected as a result of the production of artificial snow and the resulting surface runoff or increased inflow of artificial snow-melt water, as well as the operation of the accompanying infrastructure.

The quality of water in the Remiaszów stream was variable throughout the study period, particularly in the case of microbiological indicators of fecal contamination, as evidenced by a very high coefficient of variation, even higher than the one observed in the Białka river. This was most probably caused by the fact that during the study period, the surrounding environment of the stream has undergone considerable changes. For the purpose of the construction of the new ski slope, 12% of vegetation was removed [13], and intensive earthworks were conducted, followed by the changes in the amount of water in the watershed, i.e., inflow of melt water from artificial snow and treated sewage from the ski slope-related infrastructure. However, as shown by Fidelus-Orzechowska et al. [13], the environmental impact of the construction of both ski runs in the catchment of the Remiaszów stream, i.e., changes in land relief, drainage patterns or erosion rates, was smaller than in the case of most similar areas in other parts of the world. Additionally, in terms of the concentration of bacteriological indicators, this site can be classified as having water of excellent quality [19]. Moreover, the mean concentration of microbiological indicators of pollution in the Remiaszów stream was smaller than the one recorded for the Białka river (intake for the production of artificial snow). This is because along the river Białka, there are numerous sources of pollution, situated upstream of the intake site, affecting the quality in the investigated location [20].

The chemical composition and the amount of indicator bacteria in the water used for the production of artificial snow indicate that the water in the Białka river is contaminated. The prevalence of the examined microbiological indicators is characterized by a considerable variation. This was discussed in previous papers, demonstrating that waters of the Białka river are contaminated by numerous point and nonpoint sources, including surface runoff from the surrounding hills, feces of sheep grazing on the slopes but also the inflow of treated and untreated sewage both from households acting as private guesthouses and from the municipal treatment plant [20,21]. It is worth noting that Białka is the largest river draining the only high mountain range in Poland. According to Rutkowska et al. [22], the catchment to the Trybsz water gauge has an area of 202 km$^2$ and flood quantile Q (m$^3$ s$^{-1}$) of return period T = 100 years is 431.7 m$^3$/s. In the winter, water resources are scarce therefore, to ensure the proper amount of water for artificial snow production, retention reservoirs were built. Additionally, the temperature of the water in Białka flowing from the Polish and Slovak Tatras is low, approx. 5 °C and during the winter, when the air temperatures are low, the water temperature sometimes drops to 0 °C, which means that it partially freezes. This results in water resources being particularly low in winter [23]. With regard to the fact that water used for the production of artificial snow is stored in a technical reservoir prior to its use, the majority of bacteria contaminating the water will remain in the reservoir sediment [24], while its chemical composition might affect the water

in the Remiaszów stream, mostly during the spring snow-melt period, when water from the artificial snow will reach the stream.

The land cover changes, caused by the construction of the ski slope, i.e., among other things—tree cutting, soil cover violation and destruction of meadow vegetation leading to increased leaching of $K^+$, $Mg^{2+}$, $NO_3^-$ and $SO_4^{2-}$ ions, together with bacteria, i.e., total coliforms, total *E. coli*, fecal *E. coli* and mesophilic bacteria. This caused an increase in the concentration of these parameters in the water of the Remiaszów stream during the construction works. Such phenomenon of increased leaching due to changes in the land cover and disturbance of the soil cover has often been described in the literature. Many authors [25–28] indicate that the removal of vegetation, most importantly trees, decreases nutrient assimilation, particularly $NO_3^-$. The observed increase in the concentration of cations: $Ca^{2+}$, $Mg^{2+}$, $K^+$ in stream waters is also associated with the disturbance of the soil cover and the resulting increased erosion, which affects faster leaching of these compounds from the catchment [25,27,29–32]. However, over time and with the recovery of the plant cover, the ion concentrations drop to the level recorded before the land cover change [33,34]. Similarly, in studies conducted by Wang et al. [27], the chemical composition of the water had already returned to the condition recorded before the forest clearance after approximately one year after the initiation of the harvest. It was also observed that the $SO_4^{2-}$ concentrations still grow, while the level of the remaining ions decreases. Similar results were obtained by Nodvin et al. [33], who associated the results with acidification and sulfate adsorption processes in the soil.

Construction of the new ski slope and, in particular, the production of artificial snow made of water from the river Białka, resulted in increased values of pH, EC, TDS and the concentrations of $Ca^{2+}$, $Mg^{2+}$ and $HCO_3^-$ in the water of the Remiaszów stream. This is caused by the supply of melt water from artificial snow, which is much more mineralized than natural atmospheric precipitation. On the other hand, the concentrations of $Na^+$, $K^+$, $Cl^-$ and $NO_3^-$ returned to the ones observed before the construction of the slope and violation of the plant–soil cover. This may evidence their retention in the catchment by the reviving vegetation on the mentioned slope.

The disturbance of the soil cover also increases the amount of suspension in the water, resulting in elevated numbers of bacterial indicators of poor water quality. It was demonstrated by numerous authors that the number of bacteria in water is significantly associated with the amount of suspended solids [35,36]. Another factor affecting the increased number of bacteria in water can be related to sheep grazing on the ski slope. For instance, Martinsen et al. [37] demonstrated that on the slopes where sheep are grazed, increased leaching of coliform bacteria occurs, whereas this effect was not observed in the case of nitrogen compounds. Additionally, the fresh feces of grazing animals is one of the two main non-human sources of waterborne fecal indicator microorganisms [38]. Another important issue, concerning the potential contamination of surface water with sheep feces-derived bacteria, is that both *Escherichia coli* and fecal enterococci not only survive in animal feces even for c.a. 60 days after deposition, but also their number increases significantly over the first three weeks, with quite a considerable impact on the quality of surface water in the close proximity to the sites where sheep are grazed [39].

Finally, the ability of natural streams for self-purification is yet another factor affecting the observed decrease in the concentration of the examined parameters of water pollution. The self-purification ability of stream waters is the combination of various physical, chemical and biological processes, including sedimentation, adsorption, biological mineralization or assimilation [40]. The rate of the self-purification process is affected by the temperature, solar radiation, dissolved oxygen, flow rate, water depth, pH, the content of toxic compounds, sediment load as well as human activity and living organisms in the river [41]. What also should be mentioned here is that it would be valuable to refer the observed numbers of bacterial indicators of water quality and concentrations of physico-chemical parameters to the streamflow data in order to obtain the loads of the examined parameters.

Thus, evaluating load differences could allow a fuller assessment of the net effect of the development of a ski resort on water quality.

What needs to be remembered, is that—apart from its undisputable impact on the natural environment—mountain tourism also has many economic benefits [42]. With regard to the fact that the mountain regions in Poland are mainly rural, economically under-developed areas, mountain tourism, and especially winter tourism resulting from ski resort development, is an important source of jobs [14]. Therefore, the creation of new and the development of existing ski resorts, including the construction of new ski runs, slopes, lifts and related infrastructure, is inevitable in the mountain regions in Poland. An important aspect of this development is to bear in mind that maintaining natural resources in their best shape is also essential for the future development of mountain areas, as a clean environment, including stream waters, is the best showcase of mountain regions, attracting crowds of tourists.

## 5. Conclusions

As the development of ski resorts in the mountain regions in Poland is unavoidable and brings clear benefits, it is important to assess the impact of newly constructed investments on local water resources.

Our study shows that the processes involved in the construction of the new ski slope deteriorated the quality of water in the nearby stream both in terms of physico-chemical parameters and bacteriological indicators of water quality. Undoubtedly, the operation of the newly constructed ski slope also provides a considerable change in the environmental conditions, with numerous aspects. One of the most important of these aspects is the production of artificial snow, which changes the amount of water in the watershed and provides higher concentrations of ions and bacteriological contamination. Additionally, the change of the land use (from forests to ski slopes, which in spring changes into meadows where sheep are grazed) considerably affects leaching of ions from the soil cover and adds even more bacteriological contaminants to the surrounding streams.

However, most importantly—when the construction works in the studied region were finished, i.e., during the operation and maintenance of the ski slope—the properties of the nearby stream returned to the condition observed before the ski slope construction. This could indicate that the analyzed stream has quite large self-purification abilities and could also be the result of revegetation.

Such studies, involving the assessment of the local surface water quality and factors affecting it, covering the period before, during and after the construction of new ski run runs, should be conducted in every case of such investments.

**Supplementary Materials:** The following supporting information can be downloaded at: https://www.mdpi.com/article/10.3390/app13020763/s1, Table S1: Raw data of microbiological and physico-chemical indicators of water quality assessed in this study.

**Author Contributions:** Conceptualization, A.L.-B. and A.B.; methodology, A.L.-B., A.B. and M.Ż.; software, M.Ż.; validation, M.Ż.; formal analysis, A.B.; investigation, A.L.-B. and A.B.; resources, A.L.-B. and M.Ż.; data curation, M.Ż.; writing—original draft preparation, A.L.-B.; writing—review and editing, A.B. and M.Ż.; visualization, A.B.; supervision, M.Ż.; project administration, A.L.-B. and M.Ż.; funding acquisition, A.L.-B. and M.Ż. All authors have read and agreed to the published version of the manuscript.

**Funding:** This study was funded by the statutory measures of the University of Agriculture in Krakow within grant nos. BM 4149/2014, DS 3102/KM and DS 3158/KM, as well as within the measures of the project "Hydrochemical and Hydrological Monitoring of the Białka Subcatchments in the Neighbourhood of Kotelnica" (K/KDU/000153 and K/KDU/000297 and the Jagiellonian University, head of the project—Mirosław Żelazny).

**Institutional Review Board Statement:** Not applicable.

**Informed Consent Statement:** Not applicable.

**Data Availability Statement:** The authors declare that the data supporting the findings of this study are available within the article.

**Acknowledgments:** The authors would like to thank Łukasz Jelonkiewicz for his assistance in hydrochemical analyses.

**Conflicts of Interest:** The authors declare no conflict of interest.

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
