# Peer review of "Impact of Construction and Functioning of a Newly Built Ski Slope on the Quality of Nearby Stream Water"

_applsci, doi:10.3390/app13020763_

Round 1

Reviewer 1 Report

Reviewer

MDPI – Applied science

Manuscript Number: applsci-2134673

Title: « Impact of Construction and Functioning of a newly built Ski Slope on the Quality of nearby Stream Water ».

line 13-24 The abstract lacks the most important result obtained in this article. Now the abstract is a listing of facts about the negative impact of slope construction on physico-chemical and biological properties. The main conclusion must be confirmed by digital material (relative or absolute units of measurement).

line 24 Abstract: what is the further direction of the research results

In conclusion, I would also like to see the basis for the conclusion based on the results with digital data.

There are a lot (more than 60%) of sources older than 2012 in the list of references.

Author Response

Dear Reviewer,

Thank you for the review of our manuscript.

Below you you can find replies to the remarks:

1. The abstract was improved and more information, as well as further suggestions for applying the presented research were added.

2. Supplementary data sheet was provided with detailed numerical values supporting the main conclusions of the study.

3. Ten references have been changed into newer ones. Some of the references that were older than 2012 are among the most relevant and most up-to-date studies that concern the issue that the study deals with (e.g. the papers referring to the impact of mountain tourism on the environment or biogeochemical processes occurring in the water).

In the attachment we uploaded the corrected and, hopefully, corrected version of the manuscript.

Reviewer 2 Report

The paper “Impact of Construction and Functioning of a newly built Ski Slope on the Quality of nearby Stream Water” is an interesting manuscript that analyse the impact of the construction and operation of a new ski slope with its infrastructure and artificial snow production, on the quality of nearby stream waters. The paper is well written. The results are well presented and the discussion is exhaustive. The paper can be accepted for publication on Applied Sciences 

Author Response

Dear Reviewer,

thank you very much for a positive review of our manuscript.

In the attachment we uploaded the corrected and, hopefully, improved version of the manuscript.

With kind regards,

Anna Lenart-Boroń

Reviewer 3 Report

Generally, this is a very interesting study in which we can assess the direct impact on water quality after a construction project. Also, it gives us a perspective on how the environment mitigate and comes to normality after a project and the importance of a monitoring program on areas impacted.  The statistic is good and in general the study is very good.  All the data is relevant.

To be specific, you have compared date from 3 years.  This provided a baseline to use as a reference for your conclusions.  Also we could see the anthropogenic impact on the stream and how with little or no human interaction, how the parameters, after the impact (construction) started to stabilize.  In addition, you took advantage of the data precipitation provided toward the water quality parameters, therefore, you could assess the long-term impact of natural and anthropogenic factors toward the water quality of the stream.  Not many research projects take all those factors into consideration and the majority just acknowledge those as confounding variables but in a more abstract manner.  Another important factor was the statistic (ANOVA) you used to compare, natural and anthropogenic impacts on the stream on Physical, Chemical and Biological (microbiological) variables. 

All of this, in my view, make the study robust and one that it can be use as example for other similar studies.  This will give us more reliable data for policies on water quality, permits, environmental health and sustainability.

Author Response

Dear Reviewer,

thank you very much for a thorough, but at the same time positive review of our manuscript.

In the attachment we are sending the corrected and, hopefully, improved version.

With kind regards,

Anna Lenart-Boroń

Reviewer 4 Report

The paper „Impact of Construction and Functioning of a newly built Schi Slope on the Quality of nearby StreamWater” is current and very well structurate.

The authors highlighted very well the impact of the construction and operation of a ski slope with infrastructure and artificial snow production, on the quality of nearby stream waters.

Olso, the authors presented very well  that the processes involved in the construction of the new ski slope deteriorated the quality of water in the nearby stream both in terms of physicochemical parameters and bacteriological indicators of water quality.

I propose to publish the paper in present form.

Author Response

(The authors gave the same response as above.)
